# A third generation of radical fluorinating agents based on N-fluoro-N-arylsulfonamides

Daniel Meyer [1], Harish Jangra[2], Fabian Walther[1], Hendrik Zipse [2] & Philippe Renaud [1]

Radical fluorination has been known for a long time, but synthetic applications were severely limited by the hazardous nature of the first generation of reagents such as $F_2$ and the strongly electrophilic nature of the second generation of reagents such as N-fluorobenzenesulfonimide (NFSI) and Selecfluor®. Here, we report the preparation, use and properties of N-fluoro-N-arylsulfonamides (NFASs), a class of fluorinating reagents suitable for radical fluorination under mild conditions. Their N–F bond dissociation energies (BDE) are 30–45 kJ mol$^{-1}$ lower than the N–F BDE of the reagents of the second generation. This favors clean radical fluorination processes over undesired side reactions. The utility of NFASs is demonstrated by a metal-free radical hydrofluorination of alkenes including an efficient remote C–H fluorination via a 1,5-hydrogen atom transfer. NFASs have the potential to become the reagents of choice in many radical fluorination processes.

[1] Department of Chemistry and Biochemistry, University of Bern, Freiestrasse 3, 3012 Bern, Switzerland. [2] Department of Chemistry, LMU München, Butenandtstrasse 5-13, 81377 München, Germany. Correspondence and requests for materials should be addressed to H.Z. (email: zipse@cup.uni-muenchen.de) or to P.R. (email: philippe.renaud@dcb.unibe.ch)

The introduction of fluorine atoms into organic molecules significantly changes their physical, chemical, and biological properties, and is therefore very attractive for the preparation of innovative materials, agrochemicals, and pharmaceuticals[1–3]. Moreover, [18]F-labeled organic compounds are of high clinical interest as contrast agents for positron emission tomography (PET)[4–6]. This situation has created a strong demand for efficient fluorination techniques. In the last 30 years, the introduction of fluorine atoms using nucleophilic and electrophilic reagents has led to remarkable advances. Radical fluorination has been known for a long time, but synthetic applications were severely limited by the hazardous nature of the first generation of reagents (Fig. 1a) such as F₂[7], hypofluorites (ROF)[8], and XeF₂[9]. Recently, a second generation of reagents, initially developed and optimized for electrophilic fluorination, changed dramatically that picture and radical fluorination is becoming an essential tool for selective fluorination under mild conditions (Fig. 1b)[10–13]. Sammis and co-workers[14] proposed in 2012 that *N*-fluorobenzenesulfonimide (NFSI), Selectfluor®, and *N*-fluoropyridinium salts (NFPY), due to their low N–F bond dissociation energies (BDE), may be used for radical fluorination. This hypothesis was confirmed by the description of a radical fluorinative decarboxylation of *tert*-butyl peresters (Fig. 1b)[14] and 2-aryloxy carboxylic acids using NFSI[15] as a source of fluorine atom. NFSI was also used by Zhang et al.[16] for the copper-catalyzed aminofluorination of styrene, by Britton and co-workers[17] for a tetra-*n*-butylammonium decatungstate-catalyzed C(sp³)–H bond fluorination, and by Lectka and co-workers[18] for the aminofluorination of cyclopropanes. Following the work of Li on the Ag(I)-catalyzed fluorodecarboxylation with

Selectfluor®[19,20], this reagent became the most common reagent for radical fluorination processes[11]. Using this reagent, the decarboxylative fluorination[21–24] has been thoroughly investigated and very recently the fluorination of tertiary alkyl halides was reported[25]. Interestingly, the fluorinative deboronation of alkylpinacolboranes and alkylboronic acids catalyzed by Ag(I) with Selectfluor® was reported by Li (Fig. 1b)[26]. Aggarwal and co-workers[27] reported that such a radical process involving Selectfluor® was a competing reaction during the electrophilic fluorination of boronate complexes. Boger and Barker[28] developed an Fe(III)/NaBH₄-mediated free radical Markovnikov hydrofluorination of unactivated alkenes with Selectfluor®. A related cobalt-catalyzed hydrofluorination reaction was reported by Hiroya and co-workers[29] using a *N*-fluoropyridine source of atomic fluorine. Groves and co-workers[30,31] developed recently an appealing manganese-catalyzed procedure for C–H fluorination process using the nucleophilic F⁻ as the fluorine source.

The second generation of radical fluorinating agents has transformed the field. However, they are often penalized by the necessity to use a transition metal catalyst and by their strong electrophilic/oxidative character. A careful look at the reaction mechanisms shows that they are frequently involved in electron transfer processes and that carbocation intermediates are generated by overoxidation processes. This was clearly demonstrated by Li and co-workers[19] for the non-catalyzed fluorinative decarboxylation of peresters with Selectfluor® in the absence of a Ag(I) catalyst. A third generation of reagents designed to work efficiently under mild radical reaction conditions without being involved in electrophilic or electron transfer processes is clearly needed[32,33]. We report here that *N*-fluoro-*N*-arylsulfonamides

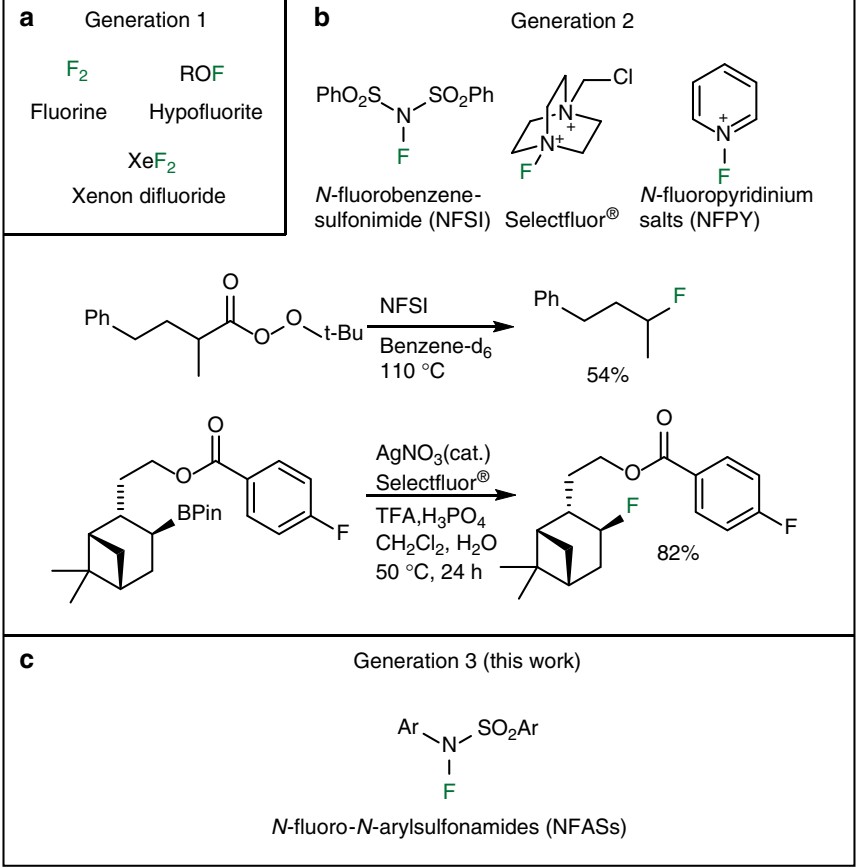

**Fig. 1** The three generations of reagents for radical fluorination. **a** Fluorination reagents of first generation. **b** Example of fluorinative decarboxylation and deboronation reactions using reagents of the second generation. **c** General structure of the *N*-fluoro-*N*-arylsulfonamides (NFASs) described in this work

(NFASs) belong to this third generation of radical fluorinating reagents (Fig. 1c). NFASs have been optimized for the catecholborane-mediated hydrofluorination of alkenes and tested in the fluorinative decarboxylation of peresters.

## Results

**Design of radical fluorinating agents.** Initial investigations of the hydrofluorination of alkenes started with the hydroboration of 1-phenyl-1-cyclohexene **1a** with catecholborane followed by reaction with Selectfluor® and NFSI as fluorinating agents (Fig. 2a). Reaction with Selectfluor® was highly exothermic and led to decomposition of the intermediate *B*-alkylcatecholborane. No trace of the fluoride **2a** was detected by GC analysis. The reaction with NFSI afforded **2a** in 15% yield. In order to suppress undesired side reactions caused by the electrophilicity of the fluorinating agents, less electrophilic N–F reagents were tested. Benzenesulfonamides **3a**–**3b** and benzamide **3c** were prepared by fluorination of the corresponding amides[34] and tested, but all three *N*-fluoroamides proved to be inefficient (yields ≤ 4%).

The disappointing results obtained with the *N*-fluoro-*N*-alkylamides **3a**–**3c** were interpreted as a consequence of a too high BDE of the N–F bonds. In order to put this hypothesis on a quantitative basis, N–F BDEs were calculated for Selectfluor®, NFSI, and **3a**–**3c** in the gas phase and in DMF solution (Fig. 2b). As in previous studies on radical stabilities of *N*-centered radicals, geometry optimizations have been performed at the (U)B3LYP/6–31G(d) level of theory[35]. Thermochemical corrections to 298.15 K have been calculated at the same level of theory using the rigid rotor/harmonic oscillator model. Improved relative energies were obtained using the (RO)B2PLYP/G3MP2Large and G3(MP2)-RAD scheme proposed by Radom and co-workers[36,37]. The stabilities for *N*-centered radicals obtained from fluoramides R$_2$N–F have been determined with reference to fluoroamine (H$_2$N–F) using the isodesmic fluorine exchange reaction shown in Eq. (1).

$$R_2N-F + \cdot NH_2 \rightarrow R_2N\cdot + F-NH_2 \quad \Delta H_{298} = RSE(R_2N\cdot)$$
(1)

$$BDE(R_2N-F) = RSE(R_2N\cdot) + BDE(H_2N-F)$$
(2)

The reaction enthalpies ($\Delta H_{298}$) obtained from Eq. (1) (commonly referred to as radical stabilization energies of the substrate radicals R$_2$N•) can be combined with the reference value for the H$_2$N–F parent system (+286.6 kJ mol$^{-1}$)[38] to obtain N–F BDE values of the fluoroamines R$_2$N–F as expressed in Eq. (2). The trends in N–F BDE values are very similar at all levels of theory and also in the gas phase and in DMF solution (see Supplementary Figs. 219–221 and Supplementary Tables 4–7). For the sake of brevity we will only discuss the results obtained at the G3(MP2)-RAD level. In DMF solution the N–F BDEs of **3a**, **3b**, and **3c** are calculated to be 263.0, 263.6, and 274.6 kJ mol$^{-1}$ (62.9, 63.0, and 65.6 kcal mol$^{-1}$), which is close to the N–F BDE in Selectfluor® (265.7 kJ mol$^{-1}$, 63.5 kcal mol$^{-1}$), but slightly higher than in NFSI (259.3 kJ mol$^{-1}$, 62.0 kcal mol$^{-1}$) (Fig. 2b). These results are in line with the fact that such *N*-alkylamidyl radicals are only weakly stabilized[35] and have been used recently for C–H chlorination, bromination, and xanthylation reactions[39–41].

In order to decrease the N–F BDE while maintaining enough polar effects to favor the fluorination of (nucleophilic) alkyl radicals, *N*-fluoro-*N*-arylsulfonamides (NFASs) **4** were investigated (Fig. 3). A solution phase N–F BDE of 222.3 kJ mol$^{-1}$ (53.1 kcal mol$^{-1}$) was calculated for *N*-Fluoro-*N*-(4-(trifluoromethyl)phenyl)benzenesulfonamide **4a**, supporting our assumption that *N*-aryl substituents should lead to lower N–F BDEs due to stabilization of the corresponding amidyl radical by delocalization onto the aromatic ring. Analyzing the impact of electron-withdrawing substituents in the anilide moiety and of electron-donating substituents in the arylsulfonyl moiety of **4a**, we find neither of these to lead to large alterations in the N–F BDE values. In fact, all N–F BDE values calculated for NFASs **4a**–**4i** cluster in the range from 220.0–226.1 kJ mol$^{-1}$ (52.6–54.0 kcal mol$^{-1}$), which is well below that for NFSI (62.0 kcal mol$^{-1}$, this value is in good accordance with the one of 63.4 kcal mol$^{-1}$ calculated recently by Xue, Cheng and co-workers)[33].

Attempts to prepare the simple *N*-fluoro-*N*-phenylbenzene-sulfonamide were not successful, presumably due to side reactions involving reaction of NFSI with the electron-rich aromatic anilide moiety. After deactivation of the anilide moiety with electron-withdrawing groups (CF$_3$, F), the NFASs **4a**–**4i** were readily prepared by fluorination of the amides upon treatment with Cs$_2$CO$_3$ and NFSI and they could be purified by flash chromatography followed by recrystallization from heptane

**Fig. 2** Initial attempts of hydrofluorination via formation of *B*-alkylcatecholboranes. **a** Hydrofluorination of 1-phenyl-1-cyclohexene (**1a**) with Selectfluor®, NFSI, and *N*-fluoro-*N*-alkylamides **3a**–**3c**. **b** Solution phase (DMF) N–F bond BDEs ($\Delta H_{sol}$) calculated at the G3(MP2)-RAD level of theory

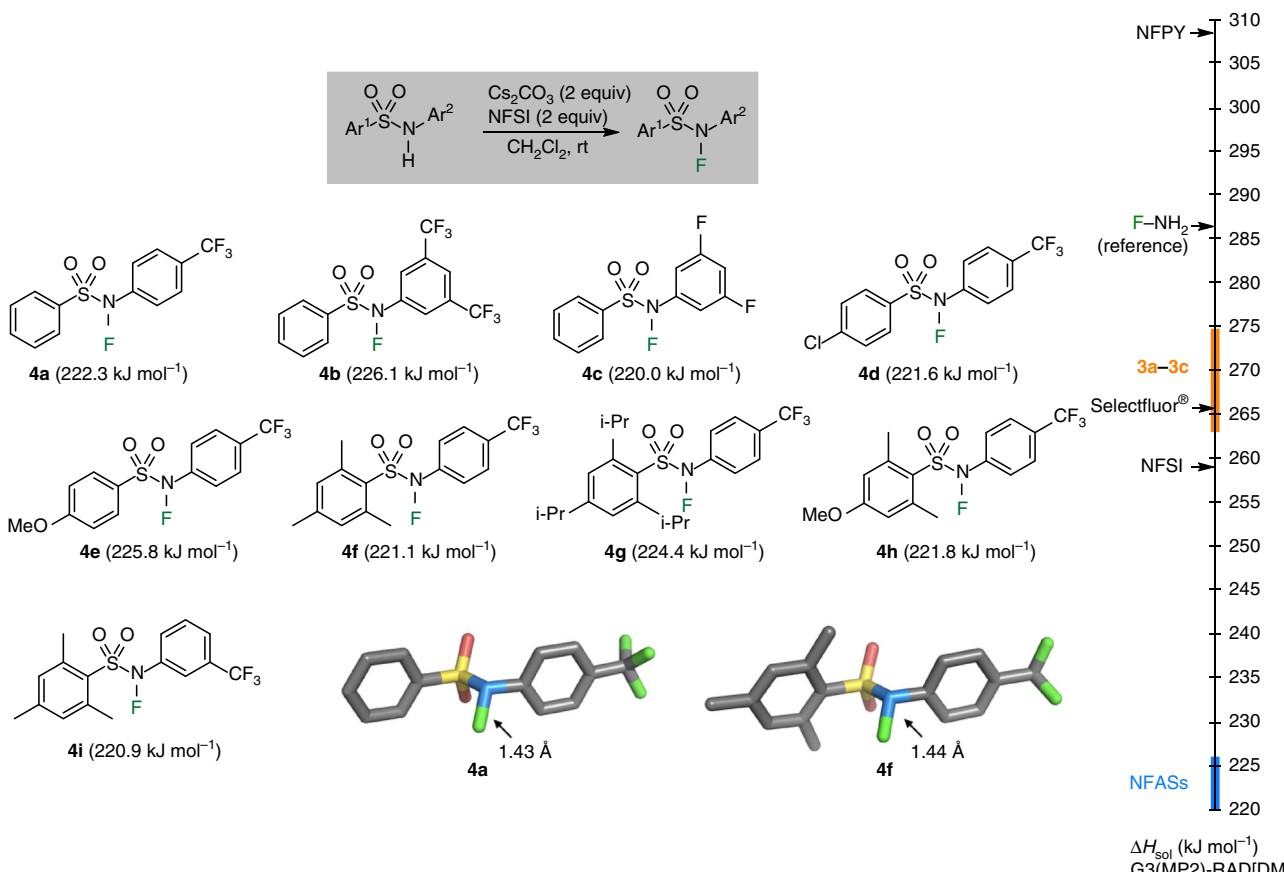

**Fig. 3** Preparation and characterization of NFASs **4a–4i**. X-ray single crystal structure of **4a** and **4f** and solution phase (DMF) N–F BDEs calculated at the G3(MP2)-RAD level of theory

(Fig. 3). The structures of **4a** and **4f** have also been determined by X-ray crystallography and are depicted in Fig. 3. The N–F bond lengths in **4a** and **4f** (1.43 and 1.44 Å, respectively) were found to be marginally longer than the N–F bond length in NFSI (1.42 Å). The structures obtained by X-ray crystallography match well with those calculated at (U)B3LYP/6–31G(d) level (see Supplementary Figs. 1, 2 and 218).

The hydrofluorination of 1-phenyl-1-cyclohexene (**1a**) with NFASs **4a–4i** was examined. Results are summarized in Table 1. The N-fluorosulfonamide **4a** was tested first using 0.1 equivalent of DTBHN as the initiator in DMF. The fluorinated product **2a** was obtained in 30% yield together with 8% of phenylcyclohexane and 10% of **1a**. Since DMF is a good hydrogen atom donor, the reaction was tested in benzene and acetonitrile[35]. However, the desired fluoroalkane **2a** was not formed in these less Lewis-basic solvents (Table 1, entries 2–3). Other solvents such as N-methylformamide, N-methyl-2-pyrrolidone, and hexamethylphosphoramide were also tested, but they provided no improvement over DMF. Using a larger amount of the radical initiator DTBHN led to a slight but reproducible increase of the yield (Table 1, entries 4–5, 45%). The other NFASs **4b–4i** were tested under the optimized reaction conditions of entry 4 (0.5 equivalent DTBHN, DMF at 80 °C). NFASs bearing a second electron-withdrawing group such as **4b–4d** gave lower yields (Table 1, entries 6–8). The other fluorinating agents **4e–4i** provided the desired fluoride **2a** in similar yields (Table 1, entries 9–13, 40–47%). For practical reasons, ease of preparation, and stability, the NFASs **4a** and **4f** were selected for further studies. All the reactions reported in

**Table 1 Hydrofluorination of 1a with N-fluoro-N-(aryl) arenesulfonamides 4a-4i**

| Entry | F-reagent yield [%][a] | Initiator (equiv) | Solvent | T (°C) | 2a |
|---|---|---|---|---|---|
| 1 | **4a** | DTBHN (0.1) | DMF | 80 | 30 |
| 2 | **4a** | DTBHN (0.1) | Benzene | 80 | – |
| 3 | **4a** | DTBHN (0.1) | CH₃CN | 80 | – |
| 4 | **4a** | DTBHN (0.5) | DMF | 80 | 45 |
| 5 | **4a** | DTBHN (1) | DMF | 80 | 45 |
| 6 | **4b** | DTBHN (0.5) | DMF | 80 | 9 |
| 7 | **4c** | DTBHN (0.5) | DMF | 80 | 23 |
| 8 | **4d** | DTBHN (0.5) | DMF | 80 | 30 |
| 9 | **4e** | DTBHN (0.5) | DMF | 80 | 41 |
| 10 | **4f** | DTBHN (0.5) | DMF | 80 | 47 |
| 11 | **4g** | DTBHN (0.5) | DMF | 80 | 43 |
| 12 | **4h** | DTBHN (0.5) | DMF | 80 | 41 |
| 13 | **4i** | DTBHN (0.5) | DMF | 80 | 40 |
| 14 | **4a** | DTBPO (0.5) | DMF | 60 | 47 |
| 15 | **4f** | DTBPO (0.5) | DMF | 60 | 51 |

[a]Yields determined by GC using n-undecane as an internal standard

Table 1, except for the bulky **4g** (entry 11), were finished in less than 10 min. Therefore, running the reaction at lower temperature was attempted. At 60 °C, the use of di-*tert*-butyl peroxyoxalate (DTBPO, easily prepared by reacting oxalyl chloride with *tert*-butyl hydroperoxide in the presence of pyridine in DMF) as an initiator[42,43] provided highly reproducible and slightly improved yields of 47% (**4a**) and 51% (**4f**) (Table 1, entries 14 and 15). The reaction is believed to be a chain process involving the reaction of the *N*-arylsulfonamidyl radical with the alkylcatecholborane to provide the desired alkyl radical. By comparison, the yield obtained with NFSI under these optimized conditions was significantly lower (29%). Beside the fluoride **2a**, small amounts of phenylcyclohexane were detected by gas

chromatography in similar quantities with all three fluorinating agents. Interestingly, the presence of the starting alkene **1a** was also observed but in significantly larger proportion with NFSI than with **4a** and **4f** (see Supplementary Table 1 and Supplementary Figs. 3–5). Since the hydroboration process takes place with complete conversion, the formation of the alkene **1a** results from undesired side reactions (see Discussion).

The scope of the metal-free hydrofluorination process was examined with non-terminal alkenes **1a**–**1i** and NFASs **4a** and **4f** (Fig. 4a). The corresponding secondary and tertiary fluorides **2a**–**2i** were isolated in 48–68% yields. In many cases, the mesitylenesulfonamide **4f** gave higher yields than the benzenesulfonamide **4a**. Cyclopentene **1b** was obtained with a good *trans*-

**Fig. 4** Hydrofluorination of non-terminal alkenes. **a** The reaction works efficiently with secondary and tertiary radicals derived from di- and trisubstituted alkenes, respectively. **b** The radical nature of the process is demonstrated by the ring-opening process observed with (+)-2-carene **1i**. **c** Preparation of the enantioenriched fluoride (–)-*trans*-**2b** from alkene **1b** is possible using (+)-isopinocampheylborane in the hydroboration step. Isolated yields are reported

selectivity (*trans/cis* 88:12). The hydrofluorination of the α-pinene- and nopol-derivatives **1c–1e** afforded **2c–2e** with high diastereoselectivities (dr ≥ 95:5). The β-citronellyl benzoate **1f** and the cholesteryl benzoate **1g** were successfully hydrofluorinated in 53% and 48% yield, respectively. Preparation of the tertiary fluoride **2h** from 1,1′-bi(cyclohexylidene) (**1h**) worked as expected (68% yield). The presence of a free radical intermediate was demonstrated with (+)-2-carene **1i** that produced the ring-opening product **2i** in 67% yield (Fig. 4b). Finally, based on our recent work on the enantioselective hydroazidation[44], a one-pot enantioselective hydrofluorination of **1b** was performed (Fig. 4c). This one-pot procedure includes a hydroboration of the alkene with (+)-IpcBH₂, conversion to the diethyl boronate, transesterification to the *B*-alkylcatecholborane and a final radical fluorination. The fluoride **2b** was isolated in 52% yield and 91:9 enantiomeric ratio.

**Kinetic data**. The rate constants for the fluorine atom transfer process between a secondary alkyl radical and NFSI, **4a** and **4f** were estimated using the cyclooct-1-en-5-yl radical clock[45–47]. The *B*-cyclooct-1-en-5-ylcatecholborane **5** was prepared by hydroboration of 1,5-cyclooctadiene and treated with the three fluorinating agents ([N–F] reagent = 1.2 M, three-fold excess) (Fig. 5a). The reaction with NFSI afforded a 75:25 mixture of the 5-fluorocyclooct-1-ene **6** and 2-fluorobicyclo[3.3.0]octane **7**. Both **4a** and **4f** afforded a nearly equimolar mixture of **6** and **7**. Based on this single concentration experiment and the published rate constant for the cyclization reaction ($k_c = 3.3 \times 10^4 \, \mathrm{s}^{-1}$ at 80 °C)[47], a rough estimation of the rate constants for fluorine transfer can be made, which for NFSI amounts to $k_F \approx 10^5 \, \mathrm{M}^{-1} \, \mathrm{s}^{-1}$ and for the two *N*-fluoro-*N*-aryl (arenesulfonamides) **4a** and **4f** to $k_F \approx 3 \times 10^4 \, \mathrm{M}^{-1} \, \mathrm{s}^{-1}$ at 80 °C (Fig. 5b). A preparative reaction was performed with **4f** on 4 mmol scale. It afforded the pure fluorides **6** (31% yield) and **7** (22%) (Fig. 5a).

**Remote fluorination**. The hydrofluorination of terminal alkenes **8a**, **8b**, and **11** was examined next (Fig. 6). The alkene **8a** gave the fluorinated product **9a** in only 7% yield together with 7% of its isomer **9a′** resulting from a radical mediated 1,5-hydrogen shift and 50% of the corresponding alkane **10a**. Running this reaction in DMF-d₇ gave **9a** (12%) and **9a′** (11%) together with 29% of the alkane **10a** with less than 5% D-incorporation. The improved

hydrofluorination/reduction ratio demonstrates that the non-deuterated DMF is probably acting as a hydrogen atom donor. However, the absence of deuterium incorporation demonstrates that other sources of hydrogen atoms are also present in the reaction mixture (including the intermediate organoborane and the fluorinating reagent itself). The methylated alkene **8b** was also investigated. The presence of the methyl group was expected to favor the hydrogen atom transfer step. Indeed, product **9b′** (30% yield) became the major fluorinated product. However, a significant amount of alkane **10b** (34%) was still produced. Based on these observations, it became clear that with suitable substrates, the radical hydrofluorination process can be used for efficient remote fluorination via 1,5-hydrogen atom transfer. A related remote fluorination process involving photoredox generated iminyl radicals has been recently reported[24]. This point is demonstrated by the hydrofluorination of the terminal alkene **11** that afforded the fluoride **12** in 68% yield with an excellent *trans* diastereoselectivity.

**Decarboxylative fluorination**. The radical fluorination ability of the NFASs **4a** and **4f** was further tested in the decarboxylative fluorination of *tert*-butyl peresters and compared with NFSI and Selectfluor® (Fig. 7). This reaction, due to its non-chain nature, is not expected to be particularly efficient and recent methods have clearly surpassed this procedure[19,23]. However, this simple reaction is very suitable to compare reagents involved in a radical mediated metal-free fluorination process. The decarboxylative fluorination of **13a** using 5 equivalents of NFSI at 110 °C (sealed tube) according to the condition of Sammis, except for the use of benzene instead of benzene-d₆, gave 3-fluoropentadecane **14a** in 5% yield together with a complex mixture of alkenes. This outcome is in line with the result of Li who ran the same reaction in benzene at 110 °C and did not observe the formation of the fluoride **14a**. All subsequent reactions were run in chlorobenzene instead of benzene to avoid the use of a sealed reaction vessel and only 2 equivalents of the fluorinating agent were used. Under these conditions, the reaction was run with NFSI, Selectfluor®, and NFASs **4a** and **4f**. NFSI provided the fluoride **14a** in 7% yield, while Selectfluor® gave only traces of **14a** (<2%, due to the high polarity of Selectfluor®, the reaction was performed in a 1:1 mixture of chlorobenzene and DMPU). Interestingly, both **4a** and **4f** gave the fluoride **14a** in moderate 48% and 47% yield. Similar

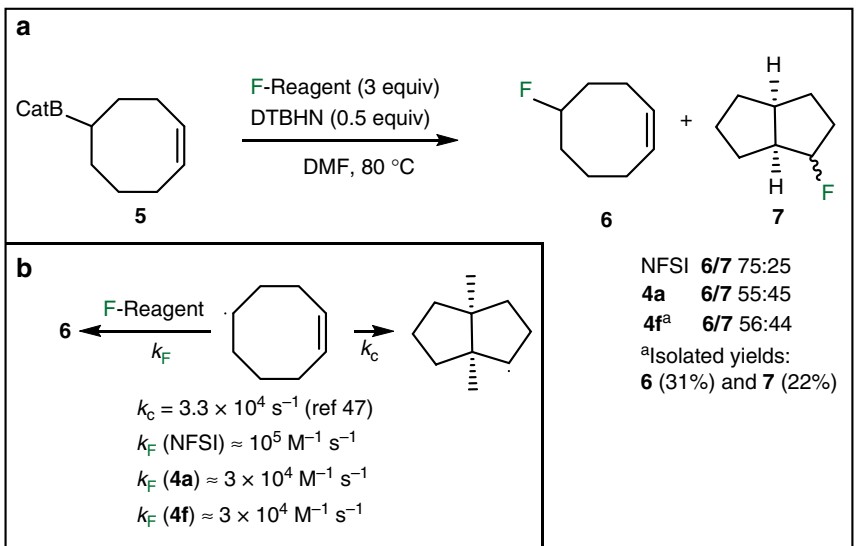

**Fig. 5** Rate constant determination using the (*Z*)-cyclooct-1-en-5-yl radical clock. **a** Fluorination of boronate **5** with NFSI, **4a** and **4f** affords mixtures of mono and bicyclic fluorides **6** and **7**. **b** Estimated rate constants for the radical fluorination

**Fig. 6** Hydrofluorination of terminal alkenes. The lower nucleophilicity of primary alkyl radical slows the direct fluorination and favors hydrogen atom abstraction processes leading to remote fluorination of unactivated C–H bonds (p-ClBz = para-chlorobenzoyl)

**Fig. 7** Decarboxylative fluorination of tert-butyl peresters. The fluorinating reagents of the second generation provide the desired fluorides in significantly lower yield than the one of the third generation due mainly to the formation of alkene side products

results were obtained with the tertiary radical derived from **13b**. Reactions with **4a** and **4f** gave **14b** in 46% and 47% yield accompanied by 35% of the alkenes. NFSI afforded only traces of the product **14b** (3%) together with larger amounts of 2-methyltetradec-2-ene and 2-methyltetradec-1-ene (64%) (see Supplementary Figs. 10 and 11). The cholic acid derivative **13c** was examined next. In that case too, NFSI (22% yield) was inferior to **4a** and **4f** (39% and 33%, respectively). Sammis and co-workers[14] reported a yield of 50% for this reaction when it was performed in deuterated acetonitrile on a 0.05 mmol scale.

**Transition states and discussion**. The higher fluorination rate observed with NFSI relative to the NFASs results is best rationalized by polar effects. The paramount importance of polar effects on the rate of radical reactions is well-established and has

been thoroughly discussed by Giese[48], Fischer and Radom[49] in their leading review articles. Polar effects have been reported to override thermodynamic effects for radical addition to alkenes[50]. Recently, Xue, Cheng and co-workers have reported that NFSI has a fluorine plus detachment (FPD) value lower than that of N-methyl-N-fluoro-p-toluenesulfonamides by 145.6 kJ mol$^{-1}$ (34.8 kcal mol$^{-1}$) in acetonitrile solution. FPD values correlated well with the reactivity of electrophilic fluorinating N–F reagents[32]. The free energy surfaces for the fluorination of the isopropyl radical in DMF solution have therefore been calculated at the (RO)B2PLYP/G3MP2large level for NFSI, **4a**, and **4f**. The calculations show slightly lower barriers for NFSI than for **4a** ($\Delta G^{\neq}_{298} = +46.1$ vs. $+51.3$ kJ mol$^{-1}$) and a somewhat higher barrier for **4f** ($\Delta G^{\neq}_{298} = +56.7$ kJ mol$^{-1}$). The transition states (TSs) for NFSI and **4f** are depicted in Fig. 8a, b, respectively. They are characterized by long C–F (2.32–2.33 Å) and short N–F

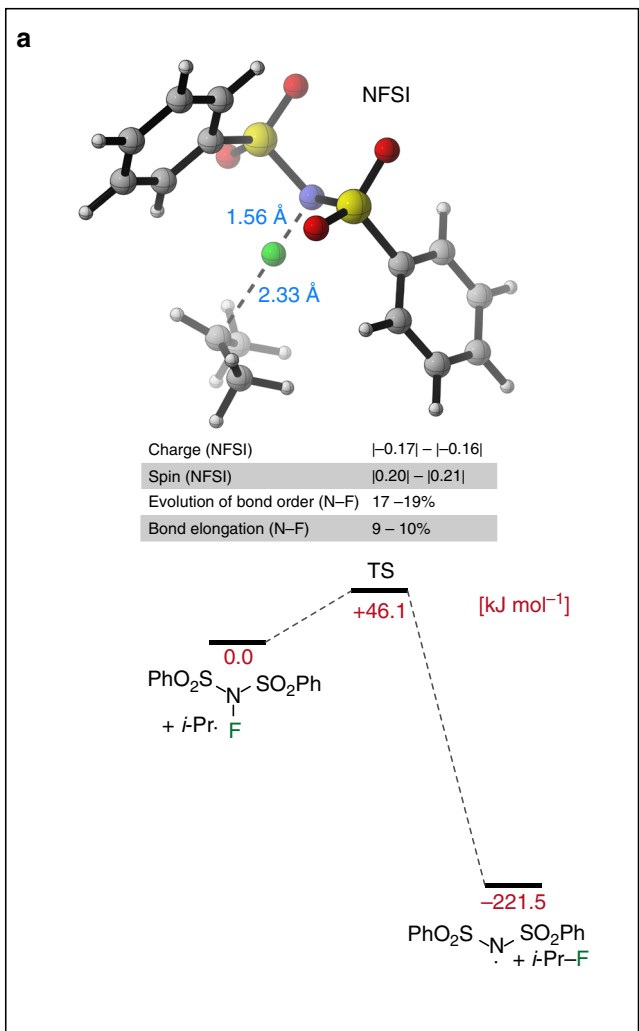

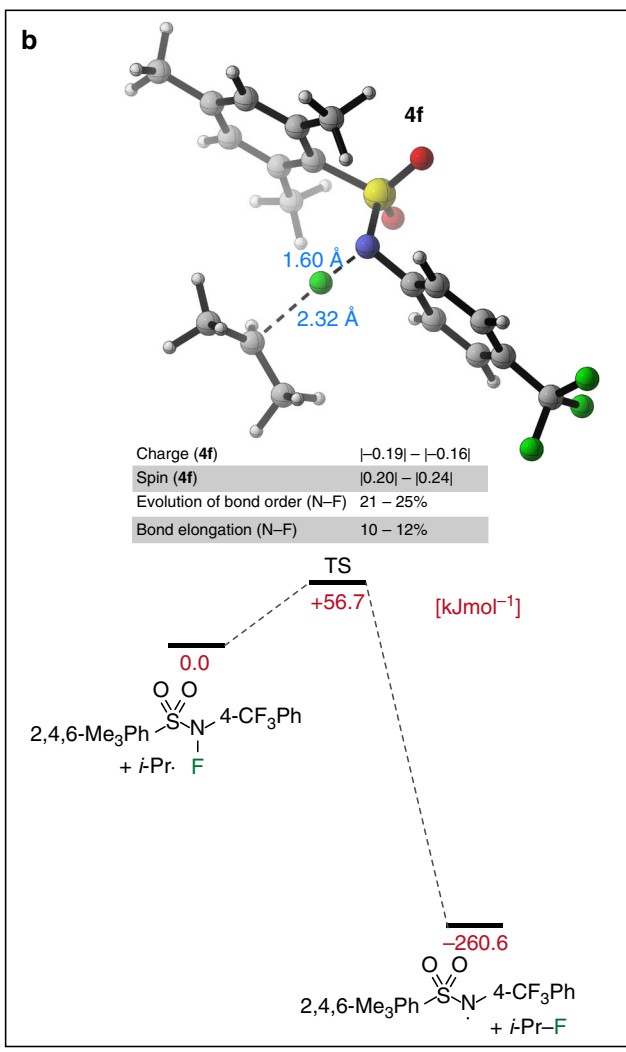

**Fig. 8** Calculated transition states for the fluorination of the isopropyl radical. **a** Free energy surfaces ($\Delta G_{\text{sol-opt}}$, in kJ mol$^{-1}$) in DMF solution for the reaction of isopropyl radical (*i*-Pr•) with NFSI and **b 4f** calculated at the (RO)B2PLYP/G3MP2Large level of theory. Distances (in Å), NPA charges and NPA spin distributions have been calculated at the SMD(DMF)/(U)B3LYP/6–31G(d) level of theory. Free energies in solution $\Delta G_{\text{sol-opt}}$ have been obtained by adding $\Delta G_{\text{solv}}$[(U)B3LYP/6–31G(d)/SMD(DMF)] to $\Delta G_{298}$[(RO)B2PLYP/G3MP2Large//SMD(DMF)/(U)B3LYP/6–31G(d)]

(1.56–1.60 Å) distances typical for very early transition states. The transition state charge distribution is very similar for all three fluorination reagents and indicates a charge transfer component of ca. 0.15–0.19e from the radical to the reagent. This charge transfer component is quite important for such an early transition state, where only 20–24% of the spin density has left the substrate isopropyl radical. Interestingly, the most significant difference between the NFSI and NFAS transition states concerns the length of the N–F bond (1.56 Å for NFSI against 1.58–1.60 Å for NFASs). In other words, the more electrophilic NFSI is able to accommodate the extra electron density caused by the charge transfer with less cleavage of the N–F bond relative to the NFASs leading to an extra stabilization of the transition state in full accordance with the polar effects aforementioned.

Both in the hydrofluorination and the decarboxylation processes, NFSI provided the desired fluorinated products in significantly lower yields than NFASs despite the observed higher rate constant for the fluorine atom transfer. For both reactions, the analysis of the crude reaction mixture showed the formation of larger quantities of alkenes for reactions involving NFSI relative to NFASs. The alkenes may result from at least three competitive processes: a single electron transfer (SET) between the fluorinating agent and the secondary alkyl radical leading to a cation followed by loss of a proton; a post fluorination acid catalyzed HF elimination; a radical cross-disproportionation process involving the alkyl radical and the imidyl radical (NFSI) or the amidyl radicals (NFASs). All these three processes are expected to be more prominent when reactions are run with NFSI relative to NFASs. Indeed, the electrophilic nature of NFSI should favor the SET process (pathway a). The HF elimination (pathway b) was experimentally found to be trigger by HF itself. The presence of HF may result from electrophilic reactions between the fluorinating agents and DMF or *tert*-butanol (hydrofluorination reaction) or traces of water (decarboxylation reaction)[51,52]. Finally, the radical cross disproportionation process (pathway c) is expected to be favored by the more reactive NFSI-derived imidyl radical over the amidyl radicals derived from NFASs. The difference of reactivity of these radicals is well-illustrated by the calculated N–H BDE for the corresponding amides (H-NFSI: BDE 454.2 kJ mol$^{-1}$; H-**4a** BDE 393.0 kJ mol$^{-1}$; H-**4f** BDE 390.6 kJ mol$^{-1}$ (see Supplementary Fig. 222 and Supplementary Table 8).

## Discussion

We have developed NFASs, a class of fluorinating reagents suitable for radical fluorination under mild conditions. The bond dissociation energies of the NFASs are 30–45 kJ mol$^{-1}$ lower than the one of NFSI and Selectfluor®. This favors smooth radical processes over side reactions caused by the electrophilic and oxidant properties of the previous generations of radical fluorinating agents. NFASs were successfully used in a metal-free hydrofluorination method involving hydroboration with catecholborane followed by a radical deborylative fluorination. By using monoisopinocampheylborane (IpcBH$_2$) in the hydroboration step, the asymmetric hydrofluorination of trisubstituted alkenes can easily be performed. Remarkably, NFASs also proved to be superior to NFSI in decarboxylative fluorination of *tert*-butyl peresters demonstrating that they are attractive reagents for a broad range of radical mediated fluorination processes. They have the potential to deeply transform the field of radical fluorination by enabling powerful transformations under milder conditions than the former generations of fluorinating agents.

## Methods

***N*-Fluoro-*N*-(4-(trifluoromethyl)phenyl)benzenesulfonamide (4a)**. To a solution of *N*-(4-(trifluoromethyl)phenyl)benzenesulfonamide (12.05 g, 40.0 mmol) in DCM (400 mL) was added Cs$_2$CO$_3$ (16.90 g, 52.0 mmol) and stirred at room temperature for 60 min. Then, NFSI (16.40 g, 52.0 mmol) was added and the mixture was allowed to stir at room temperature for 5 h. The mixture was diluted with pentane (400 mL), filtered, and concentrated. The product was purified by rapid column chromatography (heptane/TBME 85:15). Concentration of the collected chromatography fractions to a volume of 100–150 mL promoted the crystallization. The solution was stored for one night at 4 °C to yield **4a** (10.15 g, 80%) as a slightly yellow solid. $R_f$ 0.40 (heptane/TBME 9:1); m.p. 74–75 °C.

***N*-Fluoro-2,4,6-trimethyl-*N*-(4-(trifluoromethyl)phenyl)-benzenesulfonamide (4f)**. According to the procedure for **4a**, starting from *N*-(4-(trifluoromethyl)phenyl)-2,4,6-trimethylbenzenesulfonamide (17.17 g, 50.0 mmol). Crystallization at 4 °C yielded **4f** (16.16 g, 89%) as a slightly yellow solid. $R_f$ 0.55 (heptane/TBME 9:1); m.p. 116–117 °C.

**General procedure for the hydrofluorination of alkenes**. To a solution of the alkene (1.0 mmol), *N*,*N*-dimethylacetamide (14 µL, 0.15 mmol) in dry DCM (1 mL) was added dropwise catecholborane (0.23 mL, 2.2 mmol) at 0 °C. The reaction was allowed to stir at 30 °C for 16 h. The mixture was cooled to 0 °C and *t*-BuOH (0.124 mL, 1.3 mmol) was added. The reaction mixture was stirred at room temperature for 15 min, concentrated under vacuum, and the residue was dissolved in dry DMF (2 mL). DTBPO (117 mg, 0.5 mmol) and **4a** or **4f** (3.0 mmol) were added. The mixture was heated to 60 °C (preheated oil bath was used) and stirred at this temperature for 30–45 min. The crude product was purified by column chromatography.

## Data availability

Data supporting the findings of this work are available within the paper and its Supplementary Information files and from the corresponding authors on request. Source data for Supplementary Tables 9, 16 and 17 are provided as supplementary data. CCDCs 1828679 and 1828684 contain the supplementary crystallographic data for compound **4a** and **4f**, respectively. These data can be obtained free of charge from The Cambridge Crystallographic Data Centre via http://www.ccdc.cam.ac.uk/data_request/cif

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

## Acknowledgements

The Swiss National Science Foundation (Project 200020_172621) and the University of Bern are gratefully acknowledged for their support. We thank the Leibniz Super-computing Centre (www.lrz.de) for generous allocation of computational resources. We are very grateful to Lars Gnägi and Michael Hofstetter for their investigation of side reactions and to Konrad Uhlmann for his support in gas chromatography analyses.

## Author contributions

P.R. proposed the research direction and guided the project. D.M. designed and run the experimental work with the assistance of F.W. Calculations were designed by H.Z. and H. J., and performed by H.J. The manuscript was jointly written by D.M., P.R., and H.Z.

## Additional information

**Competing interests:** The authors declare no competing interests.

