## [Peer Review File · Nature Communications]

Reviewer #1 (Remarks to the Author):

Renaud, Zipse and coworkers describe in this manuscript a new class of fluorinating reagents, namely, N-fluoro-N-arylsulfonamides (NFASs). These reagents have weaker N-F bonds than NFSI or Selectfluor that are commonly used in radical fluorination reactions. This property enables a number of radical fluorination reactions to proceed smoothly with the use of NFASs, including the metal-free deboronofluorination of alkylboronates, decarboxylative fluorination of tert-butyl peresters, and even asymmetric hydrofluorination of alkenes, thus demonstrating the advantage of NFASs over NFSI or Selectfluor as radical fluorinating agents. This work is well designed and clearly supported by the experimental details in the Supporting Information. In view of the great importance of fluorine in pharmaceuticals, agrochemicals and materials science, as well as the versatility of radical fluorination in C - F bond formations, this work should have a significant impact on the way of thinking in the field of radical fluorination, and thus will be of interest to a broad audience. In addition, the manuscript is well written, and therefore I recommend its publication in Nature Commun. as is.

Reviewer #2 (Remarks to the Author):

Review for Manuscript ID: NCOMMS-18-17183-T

In this work, the authors present a new family of fluorinating reagents (N-fluoro-N-arylsulfonamides, NFAS) suitable for radical fluorination, derived from the NFSI (N-fluorobenzenesulfonimide) molecule. These molecules have lower N-F bond energies and are thus more efficient than NFSI, Selectfluor or NFPy. The article presents the genesis of this family based on theoretical calculations, their synthesis and their use in different conditions.

The article is well written and the calculations level are well suited to properly describe the electronic structure of these new molecules.

However, I do not think that the novelty brought by this work is high enough for publication in Nature Communications. Indeed, others have already proposed to use derivatives of NFSI as new fluorinating reagents that, even though may be not suitable for radical fluorination, are quite green (see for example NFSI(Me) from Shibata Green Chem 2015).

On top of this, the authors do not clearly solve the (apparent?) paradox: fluorination with NFSI is faster than with NFAS 4a and 4f, but at the same time it leads to much lower yields. Calculations of the F° transfer support the fact that fluorination with NFSI should be faster, but I could not find anything explaining why NFAS are better fluorinating reagents.

Reviewer #3 (Remarks to the Author):

The introduction of fluorine atoms into existing structures remains an important synthetic task, in view of the key importance of organofluorine compounds in pharmaceuticals and agrochemicals, in PET imaging, and in material sciences. The authors have conducted a thorough and meticulously executed (the experimental description in the SI is impeccable) study of a new family of mild fluorine atom transfer reagents allowing the introduction of a fluorine starting from organoboron and acyl peroxide precursors. Even though the yields are not perfect and room for improvement remains, this new generation of reagents is significantly superior to previous ones and more tolerant of sensitive functionality. It must be remembered that fluorination of radicals under conditions compatible with complex structures has been a longstanding problem that has been addressed with some measure of success only very recently.

In the course of this study, the authors have further adduced very valuable information regarding N-F bond strengths and, especially, rate constants for the transfer of the fluorine atom. In radical chemistry, a knowledge of even approximate rate constants is crucial for the conception and mastery of radical processes.

It would be interesting for the authors to examine the 2,4,6-trichloro (or trifluoro) analog of reagent 4f, which lacks homolytically labile benzylic hydrogens and could result in improved yields.

The manuscript is very well structured, easy to read, and clear. There are very few typing errors:

Page 1, last line, "innovative new materials" is redundant. I suggest removing "new".

Page 2, line 11 from the top, it should be "due TO their low N-F bond dissociation ..."

Page 4, last line, "Improved relative energies WERE obtained.

Page 6, first line, "Attempts to prepare N-fluor-N-phenylbenzenesulfonamide WERE not successful..."

Page 12, line 11 from the top, "and only 2 equivalents of the fluorinating agent WERE used."

Page 12, line 12 from the top, "NFSI PROVIDED..." and in line 15 on the same page ""NFSI AFFORDED..."

In reference 52, "... did not OBSERVE the formation ..."

In summary, this is an important and enduring piece of work, which will prove very useful to both academic and industrial chemists in various sectors. I wholeheartedly recommend its publication.

Response to reviewers

Reviewer #1 (Remarks to the Author):

Renaud, Zipse and coworkers describe in this manuscript a new class of fluorinating reagents, namely, N-fluoro-N-arylsulfonamides (NFASs). These reagents have weaker N-F bonds than NFSI or Selectfluor that are commonly used in radical fluorination reactions. This property enables a number of radical fluorination reactions to proceed smoothly with the use of NFASs, including the metal-free deboronofluorination of alkylboronates, decarboxylative fluorination of tert-butyl peresters, and even asymmetric hydrofluorination of alkenes, thus demonstrating the advantage of NFASs over NFSI or Selectfluor as radical fluorinating agents. This work is well designed and clearly supported by the experimental details in the Supporting Information. In view of the great importance of fluorine in pharmaceuticals, agrochemicals and materials science, as well as the versatility of radical fluorination in C - F bond formations, this work should have a significant impact on the way of thinking in the field of radical fluorination, and thus will be of interest to a broad audience. In addition, the manuscript is well written, and therefore I recommend its publication in Nature Commun. as is.

No modification required.

Reviewer #2 (Remarks to the Author):

Review for Manuscript ID: NCOMMS-18-17183-T

In this work, the authors present a new family of fluorinating reagents (N-fluoro-N-arylsulfonamides, NFAS) suitable for radical fluorination, derived from the NFSI (N-fluorobenzenesulfonimide) molecule. These molecules have lower N-F bond energies and are thus more efficient than NFSI, Selectfluor or NFPy. The article presents the genesis of this family based on theoretical calculations, their synthesis and their use in different conditions. The article is well written and the calculations level are well suited to properly describe the electronic structure of these new molecules.

However, I do not think that the novelty brought by this work is high enough for publication in Nature Communications. Indeed, others have already proposed to use derivatives of NFSI as new fluorinating reagents that, even though may be not suitable for radical fluorination, are quite green (see for example NFSI(Me) from Shibata Green Chem 2015).

The main point of our manuscript is to develop a reagent optimized for radical reactions and not for electrophilic reactions as in the work of Shibata mentioned by the referee. To the best of our knowledge, no attempt to optimize the radical fluorine atom transfer chemistry and to suppress as much as possible the electrophilic reactions has been reported. Since these two aspects are closely linked (see comment below, point 1), this task represents a major challenge, particularly if one considers the need to develop shelf-stable and easy to purify reagents.

On top of this, the authors do not clearly solve the (apparent?) paradox: fluorination with NFSI is faster than with NFAS 4a and 4f, but at the same time it leads to much lower yields. Calculations of the F^o transfer support the fact that fluorination with NFSI should be faster, but I could not find anything explaining why NFAS are better fluorinating reagents.

Thanks to the reviewer for raising this point which has to be better explained in the manuscript. Indeed, this may not be obvious for chemists not directly involved in radical chemistry to understand this apparent paradox and this point clearly deserves some explanation in the manuscript. In order to clarify this point, the following modifications have been made:

1) An explanation of the factors influencing the rate of radical reactions has been added on page 13 at the beginning of the "Transition states and discussion" part. This additional text highlights the well-established and crucial importance of polar effects on rate constants that override enthalpic effects. Leading references 56–58 were added.

2) The nature and quantity of side products has been examined by gas chromatography for the conversion of 1a to 2a. The results have been added to the supporting information (pages 36–38) and are summarized in the main text before table 1 (page 8). These results demonstrate that the hydrofluorination with NFSI affords more alkene side product as the one run with NFASs.

3) A similar study has been made for the decarboxylation reaction with perester 13b leading to 14b. In this case, the reaction with NFSI affords the fluoride 14b in low yield together with large amounts of alkenes. These results are described in the supporting information (pages 50–51) and in the main text on top of page 13.

4) Three possible pathways leading to the observed alkenes side products are described in the discussion part (page 15). All three pathways are expected to be more pregnant with NFSI than with NFAS due to the higher electrophilic character of NFSI relative to NFASs and the higher reactivity of the imidyl radical relative to the amidyl radicals. References 59 and 60 dealing with the electrophilic character of the fluorinating agents and their side reactions with solvents have been added.

Reviewer #3 (Remarks to the Author):

The introduction of fluorine atoms into existing structures remains an important synthetic task, in view of the key importance of organofluorine compounds in pharmaceuticals and agrochemicals, in PET imaging, and in material sciences. The authors have conducted a thorough and meticulously executed (the experimental description in the SI is impeccable) study of a new family of mild fluorine atom transfer reagents allowing the introduction of a fluorine starting from organoboron and acyl peroxide precursors. Even though the yields are not perfect and room for improvement remains, this new generation of reagents is significantly superior to previous ones and more tolerant of sensitive functionality. It must be remembered that fluorination of radicals under conditions compatible with complex

structures has been a longstanding problem that has been addressed with some measure of success only very recently.

In the course of this study, the authors have further adduced very valuable information regarding N-F bond strengths and, especially, rate constants for the transfer of the fluorine atom. In radical chemistry, a knowledge of even approximate rate constants is crucial for the conception and mastery of radical processes.

It would be interesting for the authors to examine the 2,4,6-trichloro (or trifluoro) analog of reagent 4f, which lacks homolytically labile benzylic hydrogens and could result in improved yields.

Excellent suggestion, however, these two compounds were found to be among the less stable NFASs we have prepared and could not be purified by a simple crystallization. Interestingly, the amount of H-atom transfer relative to F-atom transfer products remained very similar with 4a and 4f demonstrating that the methyl groups are not contributing massively to the formation of H-transfer product.

The manuscript is very well structured, easy to read, and clear. There are very few typing errors:

Page 1, last line, "innovative new materials" is redundant. I suggest removing "new".

Page 2, line 11 from the top, it should be "due TO their low N-F bond dissociation ..."

Page 4, last line, "Improved relative energies WERE obtained.

Page 6, first line, "Attempts to prepare N-fluor-N-phenylbenzenesulfonamide WERE not successful..."

Page 12, line 11 from the top, "and only 2 equivalents of the fluorinating agent WERE used."

Page 12, line 12 from the top, "NFSI PROVIDED..." and in line 15 on the same page ""NFSI AFFORDED..."

In reference 52, "... did not OBSERVE the formation ..."

All corrections were made.

In summary, this is an important and enduring piece of work, which will prove very useful to both academic and industrial chemists in various sectors. I wholeheartedly recommend its publication.

Reviewer #2 (Remarks to the Author):

In this revised version, the authors have fully answered the point I raised in my review. The additional data is very insightful.

From the other reviews, it seems that I had underestimated the novelty of this work. I thus recommend its publication in Nature Communication.

Reviewer #3 (Remarks to the Author):

In this revised version, the authors have answered adequately all the questions raised. The manuscript can now be published. I have no further remarks to add.

REVIEWERS' COMMENTS:

Reviewer #2 (Remarks to the Author):

In this revised version, the authors have fully answered the point I raised in my review. The additional data is very insightful.
From the other reviews, it seems that I had underestimated the novelty of this work. I thus recommend its publication in Nature Communication.

Reviewer #3 (Remarks to the Author):

In this revised version, the authors have answered adequately all the questions raised. The manuscript can now be published. I have no further remarks to add.